# Impact of body mass index on surgical and oncological outcomes after Hyperthermic Intraperitoneal Chemotherapy (HIPEC)

Anne-Cécile Ezanno [1,2]*, Olivier Poudevigne[1], Jean-Louis Quesada[3], Julio Abba[4], Brice Malgras[1,5], Bertrand Trilling[4,6], Pierre-Yves Sage[4], Juliette Fischer[4], Marc Pocard[7,8], Catherine Arvieux[4,9], Fatah Tidadini[4,9]*

1 Department of digestive surgery, Begin Military Teaching Hospital, Saint Mandé, France, 2 Inserm, Univ Rennes, OSS (Oncogenesis, Stress, Signaling) laboratory, UMR_S, Rennes, France, 3 Clinical Pharmacology Unit, INSERM CIC1406, Grenoble Alpes University Hospital, Grenoble, France, 4 Department of Digestive and Emergency Surgery, Grenoble Alpes University Hospital, Grenoble, France, 5 French Military health Service Academy, Ecole du Val de Grâce, Paris, France, 6 University Grenoble Alpes, UMR, CNRS, TIMC-IMAG, Grenoble, France, 7 Department of digestive surgery, La Pitié Salpêtrière Hospital, Paris, France, 8 INSERM, U965 CART unit, Paris, France, 9 Lyon Center for Innovation in Cancer, EA, Lyon 1 University, Lyon, France

* ezanno.annececile@gmail.com (ACE); FTidadini@chu-grenoble.fr (FT)

## Abstract

### Background

Complete cytoreductive surgery with Hyperthermic intraperitoneal chemotherapy (HIPEC) is the standard treatment for patients with peritoneal metastases. In this retrospective observational two-center study, we assessed the impact of patient's body mass index (BMI) on surgical and oncological outcomes.

### Methods

Between 2017 and 2021, 144 patients with peritoneal metastases (all etiologies) were included. Morbi-mortality at day-30, overall survival and free-recurrence-survival were compared according to the patients BMI. The patients were divided into 2 groups (BMI < 25, and BMI ≥ 25).

### Results

Median overall survival (OS) was 71.3 months [63–71.5], with significant differences observed between BMI groups (p = 0.015). Recurrence-free survival (RFS) averaged 26.8 months [20–35.3] and did not significantly differ between groups (p = 0.267). After stratification by histology, OS and RFS remained consistent. Cox multivariate analysis adjusted for Peritoneal Carcinomatosis Index (PCI) revealed BMI < 25 (HR = 2.53 [1.10–5.80]) and male sex (HR = 2.34 [1.11–4.92]) as predictors of poorer OS. 30-Day complication rates did not significantly differ (p = 0.094). The BMI ≥ 25

**Data availability statement:** All relevant data are within the paper and its Supporting Information files.

**Funding:** The author(s) received no specific funding for this work.

**Competing interests:** The authors have declared that no competing interests exist.

group experienced higher rates of digestive fistulas (p = 0.05) and 90-day readmissions (p = 0.007), although reintervention rates were comparable (p = 0.723).

## Conclusions

Our study suggests a potential 'obesity paradox' in the context of HIPEC procedures. Morbidity at day-30 was similar for BMI < 25, and BMI ≥ 25 patients. Readmissions at day-90 were more frequent in high-BMI group. BMI < 25 is deleteriously associated with mortality. BMI and sex were related to OS.

## Introduction

In Europe, overweight (Body Mass Index: BMI > 25 kg/m²) and obesity (Body mass Index BMI > 30 kg/m²) rates have been rising steadily for several decades. In 2023, according to data from the World Health Organization (WHO), 58.9% of adults are overweight and 29.1% of adults are obese. These rates vary from country to country, but generally, southern European countries have higher rates than northern ones [1]. The United States has the highest obesity rate in the world, with 74.8% of adults being overweight and 42.4% of adults being obese [2].

The surgical outcomes of patients in different BMI subgroups have become a widely discussed topic especially with patients undergoing major intra-abdominal cancer surgery [3–6]. The impact of obesity on postoperative outcomes of patients undergoing general elective surgery seems to be a risk factor for perioperative morbidity or mortality [4,7]. Paradoxically, overweight and obese patients undergoing nonbariatric general surgery have better outcomes than non-overweight patients, which is in contrast with the fact that a high BMI is associated with an increased risk of death in the general population [6,8,9].

Peritoneal metastases (PM) is due either to a primary peritoneal disease (mainly peritoneal pseudomyxoma and peritoneal mesothelioma), to metastasis of digestive cancer (mostly colorectal or gastric cancer) or to metastasis of ovarian cancer [10]. The management of PM continues to be a challenge due to poor outcomes. In selected patients, treatment is possible and consists of cytoreductive surgery (CRS) and hyperthermic intraperitoneal chemotherapy (HIPEC), which involves complete cytoreductive surgery followed by intraperitoneal administration of heated chemotherapy, with agent and instillation time dictated by the type of cancer. With the advent of CRS and HIPEC, 5-year overall survival (OS) has increased from less than 10% to upwards of 40–50% for PM patient [11].

The surgical and survival outcomes of patients according to the BMI has become a widely discussed topic. The impact of obesity and overweight on postoperative and outcomes of patients undergoing CRS/HIPEC has been described in a review of literature [12], which has not shown association between obesity and outcome after CRS/HIPEC. The stratification by degree of obesity shows that moderately obese patients did not have the same postoperative outcomes as severely obese patients [12–14]. OS and free-recurrence-survival (RFS) are less described in the

literature with only one publication which concluded BMI extremes, whether low or high, generally carry a poor prognosis for OS [15].

Our objective was to evaluate and compare the surgical and oncological outcomes (morbi-mortality at day-30, OS and RFS) from different BMI strata after CRS and HIPEC.

## Methods

### Study design and setting

This was a retrospective study of procedures performed at the Begin Military Teaching Hospital (HIA-B) and Grenoble-Alpes University Hospital (CHU-GA) both expert centers for HIPEC, between January 1, 2017, and December 31, 2021.

### Ethical considerations

This research is a subset of the MEC-CHIP study, a protocol registered with the Ethics Committee of the Grenoble University Hospital and listed on the Health Data Hub register, which lists trials complying with the MR004 reference methodology (n° 2205066 v 0) of the National Commission for Informatics and Liberties (CNIL). All procedures performed in this study were in accordance with the ethical standards research committee and with the Helsinki declaration.

The MEC-CHIP study focused on the cost of a hospital stay and oncological outcomes of HIPEC. All patients whose data were used in this study received a notice of information about a research project. All procedures performed in this study were in accordance with the ethical standards research committee and with the Helsinki declaration of 1975.

### Patients

All consecutive patients who had received HIPEC treatment in one of the two hospitals during the study period and who had been assessed by an interdisciplinary tumor board were included in this study. Inclusion criteria were adult with resectable PM of any etiology, eligible for HIPEC. Patients treated with cytoreduction surgery alone were excluded. Patient information for the study included age at surgery, gender, ASA score, cancer history and preoperative body mass index (BMI; kg/m$^2$). Patients in this cohort were assigned to the normal-BMI (normal weight; BMI < 25 kg/m$^2$) or the high-BMI group (overweight and obese; BMI ≥ 25 kg/m$^2$) based on their respective BMI values. In this study, open-abdomen HIPEC procedures were performed at HIA-B and closed-abdomen HIPEC procedures were performed at CHU-GA. Patient vital status was determined from medical records and from phone calls, using February 28, 2023, as the date point. The times were calculated from the dates of the HIPEC procedure. All data on signs of toxicity and postoperative complications occurring during the first 30 postoperative days (D30) were extracted from medical records and graded according to the Dindo-Clavien classification [16], with major complications being grades 3–5.

### Statistical analysis

Continuous data are presented using descriptive statistics (mean ± Standard-Deviation or median [25th; 75th percentiles]). Categorical data are presented using frequencies and percentages. Quantitative parameters were compared between groups using Mann-Whitney tests due to normality rejection. Qualitative parameters were compared between groups using the Chi-square test or Fisher's exact test as appropriate.

OS was defined as the time from the date of CRS with HIPEC until the date of death due to any cause. Survival rates were calculated using the Kaplan-Meier method. Comparison of OS between groups were analyzed using Log-Rank test and similarly for RFS. The role of confounding factors was explored using a univariate Cox proportional hazards model. Multivariate Cox proportional hazards models, with backward-stepwise selection (and a 15% threshold), were used to identify independent risk factors for OS and for RFS. A threshold of 5% was used to define the significance of

the statistical tests. No adjustment for multiplicity was applied. Statistical analyses were performed using Stata software version 16.1 (STATA, StataCorp, Texas, USA).

## Results

### Patient's characteristics

A total of 144 patients were included in our study. The median age was 59 years [52; 68] and 79 were female (54.9%). Regarding the origin of PMs, we predominantly observed cases of colorectal origin (50%) and pseudomyxoma peritonei (18.8%). Fourteen (9.7%) patients had previously undergone HIPEC. Out of the 144 patients, 75 (52.08%) had a BMI of less than 25 kg/m$^2$, while 69 (47.92%) had a BMI of 25 kg/m$^2$; or higher. Within the latter group, 9.7% were classified as obese: 6.9% had a BMI between 30 and 35 kg/m$^2$; 2.1% had a BMI between 35 and 40 kg/m$^2$ and only 0.7% had a BMI of 40 kg/m$^2$ or greater. Demographic and medical characteristics of patients are listed in Table 1.

Patients in this cohort were assigned to the normal-BMI (normal weight; BMI < 25 kg/m$^2$) or the high-BMI group (overweight and obese; BMI ≥ 25 kg/m$^2$) based on their respective BMI values. Demographic and medical characteristics of patients according to the BMI groups are listed in Table 2. The groups appear significantly different in terms of sex repartition with respectively 28 (37.3%) and 37 (53.6%) male patients in the BMI < 25 group and the BMI ≥ 25 group (p = 0.05). Frequency of patients with dyslipidaemia, hypertension and stroke/mini stroke appears to be significantly lower in the BMI < 25 group compared to the BMI ≥ 25 group (respectively p = 0.021, p = 0.014 and p = 0.023).

### Day-30 morbidity

The Day-30 morbidity was 50.7%. The rates varied significantly among the groups, with the lowest in the BMI < 25 group (44%), increasing to 52.7% in the BMI 25–29.9 group, and highest at 78.6% in the BMI ≥ 30 group. The distribution of complications according to the Clavien-Dindo classification revealed that the majority of complications were grade 1–2. However, the rate was notably higher in the highest BMI category (BMI ≥ 30) at 90.9%. Grades 3A and 3B complications were more prevalent in the middle BMI category (BMI = 25–29.9), while no Grade 3A or higher complications were reported in the highest BMI group (Table 3).

Morbidity rates at day-30 were similar between BMI < 25 and BMI ≥ 25 groups, with respectively 33 (44%) vs 40 (58%); p = 0.094. Seventeen severe complications (grade 3–5 of Clavien-Dindo classification) were observed (23.3%) without no significant differences between the 2 BMI-groups. There were significantly more digestive fistulas in the high-BMI group with 4 (5.8%) vs 0 (0%) in the normal-BMI group (p = 0.050) (Table 4).

### 90-day morbidity and readmission rate

The 90-day morbidity according Clavien-Dindo classification of grade 3 or higher was recorded, showing variations among the groups: 4% for BMI < 25, 21.8% for BMI 25–29.9, and 14.3% for BMI > 30. In detail, grade 3A complications were present exclusively in the BMI < 25 group (4%). Grade 3B complications occurred at 6.7% for BMI < 25, 10.9% for BMI 25–29.9, and were absent in the BMI > 30 group. There were no Grade 4 or 5 complications in any group (Table 5). The 90-day re-hospitalization rate was significantly different between BMI < 25 and BMI ≥ 25 groups with 4 (5.3%) vs 14 (20.3%); p = 0.007. But the 90-day reintervention wasn't different between the BMI (12% for BMI < 25 and 7% for BMI ≥ 25 groups; p = 0.723) (Table 6).

### Overall survival and recurrence-free survival

For the whole study population (n = 141, 3 patients lost to follow-up), the median OS was 71.3 months [63–71.5$^{(*)}$] all etiologies combined. The comparison of the overall survival between BMI groups was statistically significant (p = 0.017): the OS in the BMI ≥ 25 kg/m$^2$ was higher (Table 3 and Fig 1A and B).

**Table 1. Demographic, medical and hospitalization characteristics according to BMI.**

| | Population n = 144 | BMI < 25 n = 75 | BMI 25–29.9 n = 55 | BMI ≥ 30 n = 14 |
|---|---|---|---|---|
| **Age** | 59 [52;68] (26-78) | 59[54;67] (26-76) | 61[49;69] (28-78) | 55[47;57] (36-70) |
| **Sex** Male female | 65 (45.1%) 79 (54.9%) | 28 (37.3%) 47 (62.7%) | 33 (60%) 22 (40%) | 4 (28.6%) 10 (71.4%) |
| **Body Mass Index (BMI)** | 24.6 [21.6;27.5] (18.2-46.5) | 22.1 [20.2;23.1] (18.2-24.9) | 27 [26.3;28.4] (25.1-29.8) | 33.8 [31.9;35.8] (30.1-46.5) |
| ASA score > 2 | 27 (20.3%) | 13 (19.1%) | 11 (21.2%) | 3 (23.1%) |
| Active Smoker | 44 (30.6%) | 26 (34.7%) | 14 (25.5%) | 4 (28.6%) |
| Chronic Obstructive Pulmonary Disease/ Respiratory pathology | 23 (16%) | 12 (16%) | 10 (18.2%) | 1 (7.1%) |
| Dyslipidemia | 16 (11.1%) | 4 (5.3%) | 9 (16.4%) | 3 (21.4%) |
| Hypertension | 33 (22.9%) | 11 (14.7%) | 19 (34.6%) | 3 (21.4%) |
| Cardiopathy[1] | 4 (2.8%) | 1 (1.3%) | 3 (5.5%) | 0 (0%) |
| Stroke/ mini-Stroke[2] | 5 (3.5%) | 0 (0%) | 4 (7.3%) | 1 (7.1%) |
| Atrial fibrillation | 6 (4.2%) | 2 (2.7%) | 4 (7.3%) | 0 (0%) |
| Deep vein thrombosis/ pulmonary embolism | 19 (13.2%) | 8 (10.7%) | 9 (16.4%) | 2 (14.3%) |
| Diabetes | 8 (5.6%) | 3 (4.1%) | 4 (7.3%) | 1 (7.1%) |
| Renal Failure[3] | 6 (4.2%) | 4 (5.3%) | 2 (3.6%) | 0 (0%) |
| Hepatopathy[4] | 2 (1.4%) | 1 (1.3%) | 1 (1.8%) | 0 (0%) |
| Albuminemia on day CRS/HIPEC +/- 15 < 30 d/g ≥ 30 d/g | 70 (55.6%) 56 (44.4%) | 35 (57.4%) 26 (42.6%) | 24 (46.2%) 28 (53.8%) | 11 (84.6%) 2 (15.4%) |
| Abdominal Surgery | 87 (60.4%) | 45 (60%) | 33 (60%) | 9 (64.3%) |
| **Origin of PM** | | | | |
| Stomach | 11 (7.6%) | 8 (10.7%) | 2 (3.6%) | 1 (7.1%) |
| Colon or rectum | 72 (50%) | 40 (53.3%) | 29 (52.7%) | 3 (21.4%) |
| Ovary | 13 (9%) | 8 (10.7%) | 3 (5.5%) | 2 (14.3%) |
| Appendix | 15 (10.4%) | 7 (9.3%) | 7 (12.7%) | 1 (7.1%) |
| Pseudomyxoma peritonei | 27 (18.8%) | 10 (13.3%) | 12 (21.8%) | 5 (35.7%) |
| Peritoneal Mesothelium | 4 (2.8%) | 1 (1.3%) | 2 (3.6%) | 1 (7.1%) |
| Other origin | 2 (1.4%) | 1 (1.3%) | 0 (0%) | 1 (7.1%) |
| **Initial PCI** | 7 [3-16] (0-38) | 7 [3-14] (0,32) | 6 [3-14] (0,33) | 18.5 [6-24] (2,38) |
| **Characteristics of hospitalizations** | | | | |
| Length of stay (days) | 12 [10-16] 14.8 ± 10.5 (1-93) | 12 [10-16] 15.8 ± 13.5 (1-93) | 12 [10-16] 13.3 ± 4.4 (7-25) | 13 [12-16] 15.4 ± 8.8 (9-44) |
| Length of stay in intensive care unit (days) | 3 [1-5] 3.3 ± 4 (1-44) | 3 [1-5] 3.6 ± 5.2 (1-44) | 3 [1-3] 2.8 ± 1.9 (1–11) | 4.5 [2-5] 3.9 ± 1.7 (1–6) |

Mean ± standard deviation or Median [25th, 75th percentiles], (Min; Max), Number (percentage)

[1]History of coronary syndrome, heart failure, valvulopathy, [2] History of vascular accident or transient attack, [3] Chronic renal failure with GFR < 30ml/min, [4] History of viral or other hepatitis

**Table 2. Demographic, medical and hospitalization characteristics according to the BMI groups < 25 or ≥ 25 kg/m².**

| | Population n = 144 | BMI < 25 n = 75 | BMI ≥ 25 n = 69 | P-value |
|---|---|---|---|---|
| **Age** | 59 [52;68] (26-78) | 59[54;67] (26-76) | 59[49;69] (28-78) | 0.481[4] |
| **Sex** Male female | 65 (45.1%) 79 (54.9%) | 28 (37.3%) 47 (62.7%) | 37 (53.6%) 32 (46.4%) | **0.050[1]** |
| **Body Mass Index (BMI)** | 24.6 [21.6;27.5] (18.2-46.5) | 22.1 [20.2;23.1] (18.2-24.9) | 27.5 [26.3;29.7] (25.1-46.5) | **≤0.001[4]** |
| ASA score > 2 | 27 (20.3%) | 13 (19.1%) | 14 (21.5%) | 0.729[1] |
| Active Smoker | 44 (30.6%) | 26 (34.7%) | 18 (26.1%) | 0.264[1] |
| Chronic Obstructive Pulmonary Disease/ Respiratory pathology | 23 (16%) | 12 (16%) | 11 (15.9%) | 0.992[1] |
| Dyslipidemia | 16 (11.1%) | 4 (5.3%) | 12 (17.4%) | **0.021[1]** |
| Hypertension | 33 (22.9%) | 11 (14.7%) | 22 (31.9%) | **0.014[1]** |
| Cardiopathy | 4 (2.8%) | 1 (1.3%) | 3 (4.4%) | 0.350[2] |
| Stroke/ mini-Stroke | 5 (3.5%) | 0 (0%) | 5 (7.3%) | **0.023[2]** |
| Atrial fibrillation | 6 (4.2%) | 2 (2.7%) | 4 (5.8%) | 0.427[2] |
| Deep vein thrombosis/ pulmonary embolism | 19 (13.2%) | 8 (10.7%) | 11 (15.9%) | 0.350[1] |
| Diabetes | 8 (5.6%) | 3 (4.1%) | 5 (7.3%) | 0.485[2] |
| Renal Failure | 6 (4.2%) | 4 (5.3%) | 2 (2.9%) | 0.683[2] |
| Hepatopathy | 2 (1.4%) | 1 (1.3%) | 1 (1.5%) | 1.000[2] |
| Albuminemia on day CRS/HIPEC +/- 15 *< 30 d/g* *≥ 30 d/g* | 70 (55.6%) 56 (44.4%) | 35 (57.4%) 26 (42.6%) | 35 (53.9%) 30 (46.2%) | 0.690[1] |
| Abdominal Surgery | 87 (60.4%) | 45 (60%) | 42 (60.9%) | 0.915[1] |
| **Neoadjuvant Chemotherapy** | 96 (66.7%) | 55 (73.3%) | 41 (59.4%) | 0.077[1] |
| Avastin Yes/No | 22 (22.2%) | 10 (17.9%) | 12 (27.9%) | 0.233[1] |
| **Adjuvant Chemotherapy** | 85 (59%) | 44 (58.7%) | 41 (59.4%) | 0.927[1] |
| **Cancer history** | | | | |
| Synchronous PM | 69 (63.9%) | 40 (67.8%) | 29 (59.2%) | 0.354[1] |
| Metachronous PM | 39 (36.1%) | 19 (32.2%) | 20 (40.8%) | |
| Not Applicable (Primitive cancer) | 36 (25%) | 16 (21.3%) | 20 (29%) | 0.289[1] |
| **Initial PCI** | 7 [3-16] (0-38) | 7 [3-14] (0,32) | 7 [4-17] (0,38) | 0.209[4] |
| **Origin of PM** | | | | |
| Stomach | 11 (7.6%) | 8 (10.7%) | 3 (4.4%) | 0.154[1] |
| Colon or rectum | 72 (50%) | 40 (53.3%) | 32 (46.4%) | 0.404[1] |
| Ovary | 13 (9%) | 8 (10.7%) | 5 (7.3%) | 0.474[1] |
| Appendix | 15 (10.4%) | 7 (9.3%) | 8 (11.6%) | 0.657[1] |
| pseudomyxoma peritonei | 27 (18.8%) | 10 (13.3%) | 17 (24.6%) | 0.083[1] |
| Peritoneal Mesothelium | 4 (2.8%) | 1 (1.3%) | 3 (4.4%) | 0.350[2] |
| Other origin | 2 (1.4%) | 1 (1.3%) | 1 (1.5%) | 1[2] |
| **Characteristics of hospitalizations** | | | | |
| Length of stay (days) | 12 [10-16] 14.8 ± 10.5 (1-93) | 12 [10-16] 15.8 ± 13.5 (1-93) | 12 [10-16] 13.7 ± 5.6 (7-44) | 0.609[4] |
| Length of stay in intensive care unit (days) | 3 [1-5] 3.3 ± 4 (1-44) | 3 [1-5] 3.6 ± 5.2 (1-44) | 3 [1-4] 3 ± 1.9 (1–11) | 0.844[4] |

*(Continued)*

**Table 2.** (Continued)

|  | Population n = 144 | BMI < 25 n = 75 | BMI ≥ 25 n = 69 | P-value |
|---|---|---|---|---|
| **Characteristics of HIPEC** |  |  |  |  |
| Type of HIPEC |  |  |  |  |
| Open | 70 (48.6%) | 35 (46.7%) | 35 (50.7%) | 0.626[1] |
| Closed | 74 (51.4%) | 40 (53.3%) | 34 (49.3%) |  |
| Total time of operating room occupancy (min) | 501 [436-579] 516 ± 122 (266-961) | 493 [425-543] 495 ± 99 (302-764) | 503 [458-611] 539 ± 140 (266-961) | 0.071[4] |
| HIPEC duration (min) 30 45 60 90 | 30 (20.8%) 45 (31.3%) 5 (3.5%) 64 (44.4%) | 18 (24%) 20 (26.7%) 1 (1.3%) 36 (48%) | 12 (17.4%) 25 (36.2%) 4 (5.8%) 28 (40.6%) | 0.245[2] |

Mean ± standard deviation or Median [25th, 75th percentiles], (Min; Max), Number(percentage)

Chi-2 test [1], Fisher-exact test [2], Student t-test [3], Mann-Whitney test [4]

**Table 3.** Postoperative complications at day 30 according to BMI.

|  | Population n = 144 | BMI < 25 n = 75 | BMI 25–29.9 n = 55 | BMI ≥ 30 n = 14 |
|---|---|---|---|---|
| **D30 morbidity** | 73 (50.7%) | 33 (44%) | 29 (52.7%) | 11 (78.6%) |
| **Clavien-Dindo complication** |  |  |  |  |
| Grade 1–2 (n = 56) | 56 (76.7%) | 24 (72.7%) | 22 (75.9%) | 10 (90.9%) |
| Grade 3A (n = 4) | 4 (5.5%) | 3 (9.1%) | 1 (3.5%) | 0 (0%) |
| Grade 3B (n = 10) | 10 (13.7%) | 3 (9.1%) | 6 (20.7%) | 1 (9.1%) |
| Grade 4 (n = 3) | 3 (4.1%) | 3 (9.1%) | 0 (0%) | 0 (0%) |
| Grade 5 (n = 0) | 0 (0%) | 0 (0%) | 0 (0%) | 0 (0%) |
| **Type of postoperative complication** |  |  |  |  |
| Hemorrhagic complications (haemoperitoneum, abdominal wall hematoma) | 7 (4.9%) | 3 (4%) | 2 (3.6%) | 2 (14.3%) |
| Pulmonary complications (Pneumopathy or similar) | 9 (6.3%) | 5 (6.7%) | 4 (7.3%) | 0 (0%) |
| Complication in resumption of transit (ileus, gastroparesis) | 29 (20.1%) | 15 (20%) | 12 (21.8%) | 2 (14.3%) |
| Cardiac complications | 3 (2.1%) | 1 (1.3%) | 2 (2.9%) | 0 (0%) |
| Digestive Fistula | 4 (2.8%) | 0 (0%) | 2 (3.6%) | 2 (14.3%) |
| Perforation | 2 (1.4%) | 1 (1.5%) | 0 (0%) | 1 (7.1%) |
| Pancreatic or biliary fistula | 6 (4.2%) | 3 (4%) | 2 (3.6%) | 1 (7.1%) |
| Urological complication (urinary tract infection) | 10 (6.9%) | 5 (6.7%) | 3 (5.5%) | 2 (14.3%) |
| Renal failure | 6 (4.2%) | 2 (2.7%) | 4 (7.3%) | 0 (0%) |
| Neurological complications (Stroke/ TIA) | 0 (0%) | 0 (0%) | 0 (0%) | 0 (0%) |
| Bacteremia on PAC, VVC or VVP | 10 (6.9%) | 6 (8%) | 2 (3.6%) | 2 (14.3%) |
| Superficial abscess (wall) | 7 (4.9%) | 2 (2.7%) | 2 (3.7%) | 3 (21.4%) |
| Thromboembolic Complications (DVT, PE) | 2 (1.4%) | 1 (1.3%) | 1 (1.8%) | 0 (0%) |
| Evisceration | 5 (3.5%) | 2 (2.7%) | 3 (5.5%) | 0 (0%) |
| Other complications | 26 (18.1%) | 14 (18.7%) | 11 (20%) | 1 (7.1%) |
| Retransfer to intensive care | 3 (2.1%) | 2 (2.7%) | 0 (0%) | 1 (7.1%) |

Number (Frequency)

**Table 4. Postoperative complications at day 30 according to the BMI groups.**

| | Population n = 144 | BMI < 25 n = 75 | BMI ≥ 25 n = 69 | P value |
|---|---|---|---|---|
| *Mortality* | 0(0%) | 0(0%) | 0(0%) | – |
| *Morbidity* | 73 (50.7%) | 33 (44%) | 40 (58%) | 0.094[1] |
| **Clavien-Dindo complication** | | | | |
| Grade (1–2) | 56 (76.7%) | 24 (72.7%) | 32 (80%) | 0.464[1] |
| Grade (3–4) | 17 (23.3%) | 9 (27.3%) | 8 (20%) | |
| Grade (5) | 0 (0%) | 0 (0%) | 0 (0%) | – |
| *Type of postoperative complication* | | | | |
| Hemorrhagic complications (haemoperitoneum, abdominal wall hematoma) | 7 (4.9%) | 3 (4%) | 4 (5.8%) | 0.710[2] |
| Pulmonary complications (Pneumopathy or similar) | 9 (6.3%) | 5 (6.7%) | 4 (5.8%) | 1.000[2] |
| Complication in resumption of transit (ileus, gastroparesis) | 29 (20.1%) | 15 (20%) | 14 (20.3%) | 0.965[1] |
| Cardiac complications | 3 (2.1%) | 1 (1.3%) | 2 (2.9%) | 0.607[2] |
| Digestive Fistula | 4 (2.8%) | 0 (0%) | 4 (5.8%) | **0.050**[2] |
| Perforation | 2 (1.4%) | 1 (1.3%) | 1 (1.5%) | 1.000[2] |
| Pancreatic or biliary fistula | 6 (4.2%) | 3 (4%) | 3 (4.4%) | 1.000[2] |
| Urological complication (urinary tract infection) | 10 (6.9%) | 5 (6.7%) | 5 (7.3%) | 1.000[2] |
| Renal failure | 6 (4.2%) | 2 (2.7%) | 4 (5.8%) | 0.427[2] |
| Neurological complications (Stroke/ TIA) | 0 (0%) | 0 (0%) | 0 (0%) | – |
| Bacteremia on PAC, VVC or VVP | 10 (6.9%) | 6 (8%) | 4 (5.8%) | 0.747[2] |
| Superficial abscess (wall) | 7 (4.9%) | 2 (2.7%) | 5 (7.4%) | 0.257[2] |
| Thromboembolic Complications (DVT, PE) | 2 (1.4%) | 1 (1.3%) | 1 (1.5%) | 1.000[2] |
| Evisceration | 5 (3.5%) | 2 (2.7%) | 3 (4.4%) | 0.671[2] |
| Other complications | 26 (18.1%) | 14 (18.7%) | 12 (17.4%) | 0.842[1] |
| Retransfer to intensive care | 3 (2.1%) | 2 (2.7%) | 1 (1.5%) | 1.000[2] |

Number (Frequency), Chi-2 test [1], Fisher-exact test [2]

**Table 5. *D90 Severe complications (Dindo-Clavien 3-5)* according to BMI.**

| | Population n = 144 | BMI < 25 n = 75 | BMI 25–29.9 n = 55 | BMI > 30 n = 14 |
|---|---|---|---|---|
| *Mortality* | 0 (0%) | 0 (0%) | 0 (0%) | 0 (0%) |
| *Morbidity (ClavienDindo≥3)* | 17 (11.8%) | 3 (4%) | 12 (21.8%) | 2 (14.3%) |
| *D90 Severe complications* (Dindo-Clavien 3–5) | | | | |
| Grade 3A (n = 3) | 3 (2.1%) | 3 (4%) | 0 (0%) | 0 (0%) |
| Grade 3B (n = 10) | 11 (7.6%) | 5 (6.7%) | 6 (10.9%) | 0 (0%) |
| Grade 4 (n = 0) | 0 (0%) | 0 (0%) | 0 (0%) | 0 (0%) |
| Grade 5 (n = 0) | 0 (0%) | 0 (0%) | 0 (0%) | 0 (0%) |
| **D90 complication type** | | | | |
| Cardiac complications during HIPEC procedure | 1 (0.7%) | 0 (0%) | 1 (1.8%) | 0 (0%) |
| Digestive Fistula | 2 (1.4%) | 1 (1.3%) | 1 (1.8%) | 0 (0%) |
| Evisceration | 5 (3.5%) | 2 (2.7%) | 3 (5.5%) | 0 (0%) |
| Intra-peritoneal collection | 4 (2.8%) | 4 (5.3%) | 0 (0%) | 0 (0%) |
| Abdominal Occlusion | 1 (0.7%) | 0 (0%) | 1 (1.8%) | 0 (0%) |
| Bacteremia+ Implanted port removal | 1 (0.7%) | 1 (1.3%) | 0 (0%) | 0 (0%) |

**Table 6. Overall survival between BMI groups.**

| | Population n = 141 | BMI < 25 n = 74 | BMI 25–29.9 n = 54 | BMI > 30 n = 13 | p-value |
|---|---|---|---|---|---|
| *Median overall survival (Months)* | 71.3 [63; 71.5(*)] | 71.3 [58.5; ND] | (ND) | (ND) | 0.057(5) |
| **Survival rates at 12, 24, 36, 48, 60 months between BMI groups** | | | | | |
| *12 Months* | 96.5% [91.7; 98.5%] | 93.2% [84.5; 97.1%] | 100% [ND; ND] | 100% [ND; ND] | |
| *24 Months* | 86.7% [79.8; 91.4%] | 79.1% [67.8; 86.9%] | 94.1% [82.8; 98.1%] | 100% [ND; ND] | |
| *36 Months* | 79.7% [71.3; 85.9%] | 72.3% [59.9; 81.5%] | 88.5% [74.1; 95.1%] | 85.7% [33.4; 97.9%] | |
| *48 Months* | 75.1% [65.3; 82.5%] | – | – | – | |
| *60 Months* | 68.3% [54.7; 78.6%] | – | – | – | |

Median overall survival [95% Confidence Interval] (Months). Log-rank test (5)

(*)Value is underestimated; (ND) median is not available

Survival rate [95% Confidence Interval] (Percentage).

The median OS of patients with PM of colorectal origin was 71.3 months [58.5–71.3(*)] and OS for PM of colorectal (CCR) origin was similar in the BMI < 25 and BMI ≥ 25 groups (p = 0.394) (Fig 1C). For PM of pseudomyxoma origin no death occurred. Median RFS was 26.8 months [20–35.3], RFS was similar in the two groups according to BMI: 22.6 months [14.6–34.3] BMI < 25 group vs 30 months [22.7–46.6] in BMI ≥ 25 group; p = 0.267) (Fig 1D).

Median RFS for PM of colorectal, ovarian, appendix, gastric and mesothelioma origin was 21 months [13.2–44.3], 17.5 months [6.3–25.5], 35.5 months [22.7–62.7], 15 months [7.3–37.3] and 14.3 months [2.3–58.2] respectively. RFS for PM of colorectal origin was similar for BMI < 25 and BMI ≥ 25 (20 months [9.9–53.7] vs 21 months [12.4–46.6]; p = 0.947) (fig 1E). For PM of pseudomyxoma origin, the follow-up was too short to give medians of RFS p = 0.718 (Fig 1F).

## Univariate and multivariate analysis- OS according to CCR histology

Regarding OS, the multivariate analysis, adjusted for the PCI score, identified BMI < 25 (HR = 2.01 [0.78–5.22]) and male sex (HR = 3.38 [1.20–9.54]) as predictors of poorer OS (Table 7).

Regarding the RFS among 69 patients with CCR, 2 factors emerged as significant predictors in the multivariate analysis. Notably, higher-grade complications (Dindo-Clavien III-IV) substantially increased the risk of recurrence, with a hazard ratio of 3.94 (95% CI: 1.71–9.10, p = 0.001). Additionally, the number of surgical resections performed was significantly associated with decreased RFS, where each additional resection corresponded to a higher risk, yielding an HR of 1.27 per resection (95% CI: 1.10–1.46, p = 0.001). These findings underscore the critical impact of postoperative complications and the extent of surgical intervention on the long-term outcomes of CCR patients treated with HIPEC (Table 8).

## Discussion

In this study, we compared postoperative morbidity and oncological outcomes according to BMI (> or > 25 Kg/m²) in patients undergoing CRS with HIPEC for PM of all etiologies. Our study's observation of higher OS in obese patients challenges conventional understanding and suggests a potential 'obesity paradox' in the context of HIPEC procedures. This phenomenon, where increased body mass index correlates with improved survival outcomes, has been noted in other areas of medical research, including cardiovascular health and other forms of cancer treatment [6,8,9]. This finding could influence patient management strategies and the assessment of surgical risks in obese patients.

As we did not record any death, the observed 30-day mortality was lower than reported in the largest comparative NSQIP analysis published in 2019 (overall 30-day mortality was 1.1% (95%CI, 0.6%−1.6%)) [17]. Previous studies have

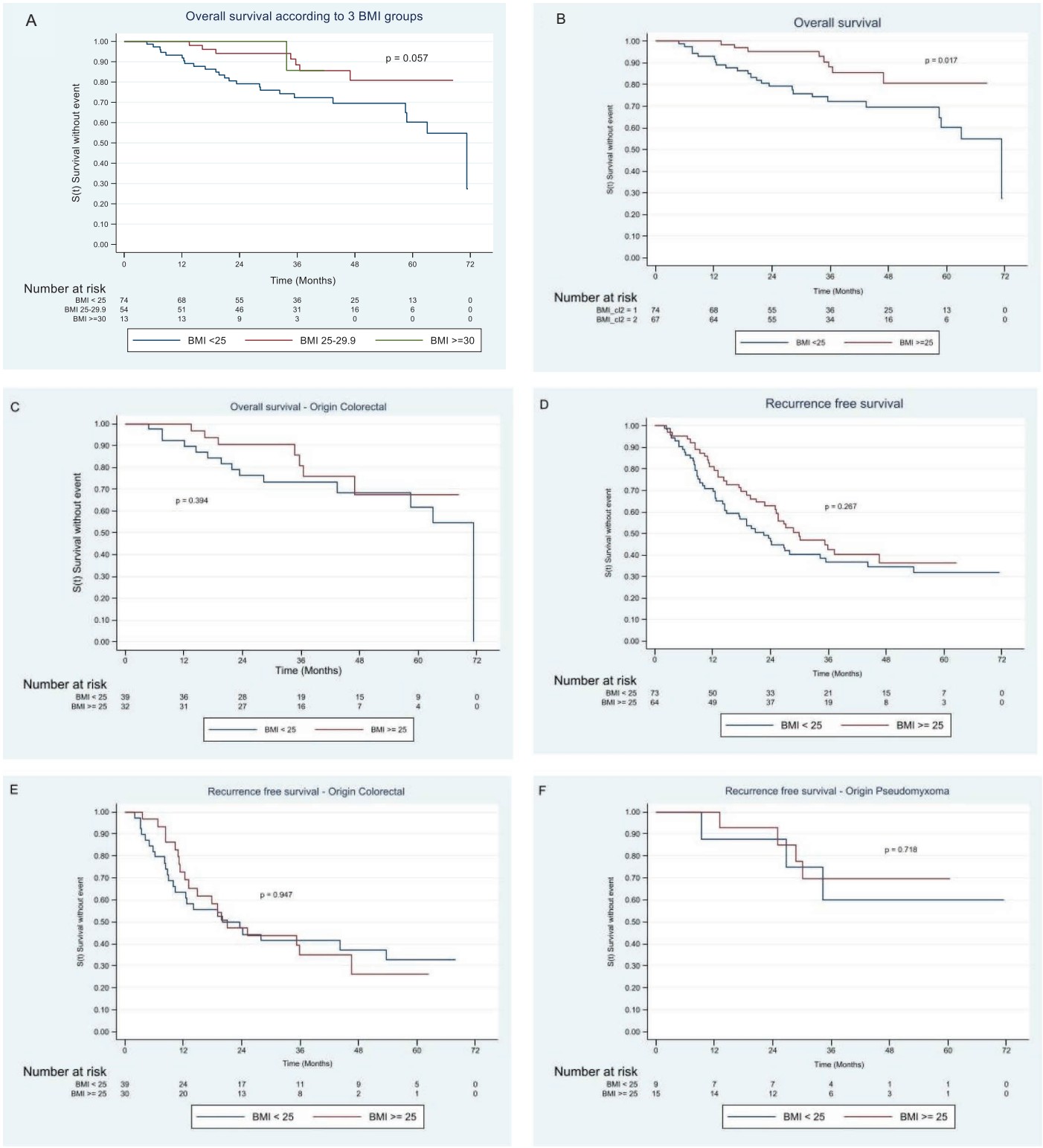

**Fig 1. Kaplan Meier overall survival curves over the entire follow-up from CRS with HIPEC.** (A) Kaplan Meier overall survival curve between BMI groups Entire cohort. (B) Kaplan Meier overall survival curve between BMI groups adjusted on HIPEC techniques. (C) Kaplan overall survival curve between BMI groups colorectal origin. (D) Kaplan Meier recurrence-free survival curve between BMI groups – entire cohort. (E) Kaplan Meier recurrence-free survival curve between BMI groups – colorectal origin. (F) Kaplan Meier recurrence-free survival curve between BMI groups – pseudomyxoma origin.

**Table 7. Univariate and multivariate Analysis – Overall survival according to CCR histology.**

| For Overall survival (n=71) | | | | | |
|---|---|---|---|---|---|
| **Factor** | | *Univariate Analysis* | | *Multivariate Analysis* | |
| | | *Hazard Ratio* | *P value* | *Hazard Ratio* | *P value* |
| BMI | < 25<br>≥ 25(Ref) | 1.49 [0.59; 3.77] | 0.397[6] | 2.01 [0.78; 5.22] | 0.150[6] |
| Type of HIPEC | *Open (Ref)*<br>*Closed* | 0.44 [0.16; 1.23] | 0.117[6] | | |
| Age | < 65 (Ref)<br>≥ 65 | 0.69 [0.26; 1.81] | 0.449[6] | | |
| Sex | *Female (Ref)*<br>*Male* | 2.84 [1.03; 7.80] | **0.044**[6] | 3.38 [1.20; 9.54] | **0.021**[6] |
| Dyslipidemia | *Yes* | 1.25 [0.29; 5.42] | 0.767[6] | | |
| Albuminemia on day CRS/HIPEC +/- 15 | < 30 d/g<br>≥ 30 d/g (Ref) | 2.36 [0.87; 6.39] | 0.091[6] | | |
| HTA | *Yes* | 1.35 [0.49; 3.73] | 0.566[6] | | |
| Stroke/ mini-Stroke | *Yes* | – | – | | |
| Previous CRC | *Yes* | – | – | | |
| Previous CRC with HIPEC | *Yes* | – | – | | |
| Previous PIPAC | *Yes* | – | – | | |
| PM Synchronous/ Metachrone | *Synchronous (Ref)*<br>*Metachrone* | 0.75 [0.30; 1.86] | 0.530[6] | | |
| ASA | *I – II (Ref)*<br>*III – IV* | 1.69 [0.63; 4.54] | 0.294[6] | | |
| PCI score | *By each point* | 1.06 [0.99; 1.12] | 0.083[6] | | |
| PCI score | ≤ 15 (Ref)<br>> 15 | 1.05 [0.24; 4.53] | 0.953[6] | | |
| *Neo-adjuvant chemotherapy* | | 3.45 [0.46; 25.93] | 0.228[6] | | |
| *Adjuvant chemotherapy* | | 0.85 [0.34; 2.12] | 0.733[6] | | |
| *Complications* | *Dindo-Clavien I – II (Ref)*<br>*Dindo-Clavien III – IV* | 0.76 [0.21; 2.82] | 0.687[6] | | |
| Intraperitoneal Chemotherapy | Mitomycin-C<br>Cisplatin<br>Doxorubicin<br>Oxaliplatin (+5-FU IV) | 0.39 [0.11; 1.37]<br>–<br>–<br>0.66 [0.21; 2.03] | 0.144[6]<br>–<br>–<br>0.468[6] | | |
| Number of resection | By resection | 1.10 [0.90; 1.35] | 0.354[6] | | |

Hazard Ratio [95% Confidence Interval]; Univariate Cox model[6]; Multivariate Cox model[7]

OS: Overall survival; HR: Hazards ratio; ASA: American Society of Anesthesiology; CRS: Cytoreductive surgery;

HIPEC: Hyperthermic intraperitoneal, chemotherapy; PM: Peritoneal metastases; PCI: Peritoneal carcinomatosis index

examined results of North American patients [12–15,17]. Our research represents the first analysis of this type of data carried out on a European population.

When comparing high-BMI patients (n=69) to normal BMI patients (n=75) (BMI ≥ 25 kg/m$^2$ vs. < 25 kg/m$^2$), there was no significant difference in length of stay in intensive care units (ICU) or hospitalization, operative time, morbidity or mortality. These findings are in agreement with previous studies, in which overweigh, and obese patients did not have increased mortality or morbidity after CRS/HIPEC [12–15]. Readmission in the late postoperative period (after 90 days) was significantly higher for the high-BMI group compared to the normal-BMI group (20.3% vs. 5.3%; p=0.007), which in line with a previous review published recently by Freudenberger [11]. We have found a higher rate of digestive fistulas in

**Table 8. Univariate and multivariate analysis – Recurrence-free survival according to CCR histology.**

| Factor | | Univariate Analysis | | Multivariate Analysis | |
|---|---|---|---|---|---|
| *for recurrence-free survival (n = 69)* | | Hazard Ratio | P value | Hazard Ratio | P value |
| BMI | < 25 <br> ≥ 25 (Ref) | 1.02 [0.56; 1.87] | 0.947[6] | 1.09 [0.49; 2.45] | 0.833[7] |
| Type of HIPEC | Open (Ref) <br> Closed | 1.03 [0.56; 1.90] | 0.930[6] | | |
| Age | < 65 (Ref) <br> ≥ 65 | 0.56 [0.28; 1.13] | 0.105[6] | | |
| Sex | Female (Ref) <br> Male | 1.44 [0.78; 2.66] | 0.243[6] | | |
| Dyslipidemia | Yes | 1.20 [0.43; 3.39] | 0.728[6] | | |
| Albuminemia on day CRS/HIPEC +/- 15 | < 30 d/g <br> ≥ 30 d/g (Ref) | 1.79 [0.92; 3.46] | 0.086[6] | | |
| HTA | Yes | 0.75 [0.33; 1.69] | 0.486[6] | | |
| Stroke/ mini-Stroke | Yes | 0.79 [0.19; 3.28] | 0.746[6] | | |
| Previous CRC | Yes | – | – | | |
| Previous CRC with HIPEC | Yes | 1.57 [0.48; 5.10] | 0.452[6] | | |
| Previous PIPAC | Yes | – | – | | |
| PM Synchronous/ Metachrone | Synchronous (Ref) <br> Metachrone | 1.31 [0.72; 2.41] | 0.379[6] | | |
| ASA | I – II (Ref) <br> III – IV | 1.59 [0.79; 3.19] | 0.193[6] | | |
| PCI score | By each point | 1.06 [1.02; 1.10] | **0.004[6]** | – | – |
| PCI score | ≤ 15 (Ref) <br> > 15 | 1.68 [0.74; 3.82] | 0.216[6] | | |
| *Neo-adjuvant chemotherapy* | Yes | 1.66 [0.65; 4.23] | 0.291[6] | | |
| *Adjuvant chemotherapy* | Yes | 1.10 [0.57; 2.14] | 0.778[6] | | |
| *Complications* | Dindo-Clavien I – II (Ref) <br> Dindo-Clavien III – IV | 3.92 [1.70; 9.04] | **0.001[6]** | 3.94 [1.71; 9.10] | **0.001[6]** |
| Intraperitoneal Chemotherapy | Mitomycin-C <br> Cisplatin <br> Doxorubicin <br> Oxaliplatin (+5-FU IV) | 0.72 [0.37; 1.39] <br> – <br> – <br> 1.28 [0.66; 2.50] | 0.330[6] <br> – <br> – <br> 0.467[6] | | |
| Number of resection | By resection | 1.27 [1.10; 1.46] | **0.001[6]** | – | – |

Hazard Ratio [95% Confidence Interval]; Univariate Cox model[6]; Multivariate Cox model[7]

RFS: Recurrence-free survival; HR: Hazards ratio; ASA: American Society of Anesthesiology; CRS: Cytoreductive surgery;

HIPEC: Hyperthermic intraperitoneal, chemotherapy; PM: Peritoneal metastases; PCI: Peritoneal carcinomatosis index

the high-BMI group (p = 0.050); which, to our knowledge, has never been found in previous studies. While other studies reported that obesity increased the risk for perioperative surgical site infections [4], our surgical site infection rate was not significantly different between the two groups (p = 0.257). A potential explanation is the BMI group configuration of our study which included not only obese patients but overweight patients as well.

For the whole study population, the median OS and RFS from CRS with HIPEC were 71.3 months and 26.8 months respectively. Comparing OS between BMI groups with a log-rank test does appear statistically significant (p = 0.017), which is not in line with the literature which did not reach statistical significance [11,13,15]. Multivariate analysis of OS identified high-BMI group as a protective factor with a reduction of 60% in the risk of death. The "obesity paradox"

describes the fact that obesity increases the risk of obesity-related disease but also is paradoxically associated with increased survival in patients with these diagnoses. This phenomenon has been observed in chronic diseases (coronary artery and end-stage kidney diseases, and heart failure), acute conditions (acute respiratory distress syndrome, sepsis and pneumonia) and critical illness in general [6,8,9,18]. However, this paradox remains a contentious issue in the oncological and surgical fields.

This study provides a detailed examination of the influence of BMI on surgical and oncological outcomes specifically in patients with colorectal cancer undergoing CRS+HIPEC. Our results indicated a median OS of 71.3 months across all etiologies, with a significant association between higher BMI (≥25 kg/m^2) and improved OS (p = 0.017), a phenomenon aligned with the 'obesity paradox' observed in other chronic conditions [19,20]. Notably, for patients with CRC origin, OS was similarly prolonged across BMI categories (p = 0.394), suggesting that BMI's protective effect might extend specifically within this subgroup. The observed protective effect of obesity in survival rates post-HIPEC surgery underscores the need to reconsider nutritional and metabolic factors as part of pre-surgical assessment and care. These insights could lead to tailored treatment plans that better accommodate the unique metabolic profiles of obese patients, potentially improving outcomes. In terms of RFS, our findings were consistent across BMI groups (22.6 months for BMI < 25 vs. 30 months for BMI ≥ 25; p = 0.267), indicating that BMI does not significantly affect the recurrence rates post-HIPEC in CRC patients. This is consistent with other studies suggesting that factors beyond BMI, such as tumor biology and treatment specifics, play more crucial roles in recurrence patterns [21,22]. Furthermore, multivariate analysis adjusted for the Peritoneal Cancer Index (PCI) identified being male and having a lower BMI (<25) as predictors of poorer OS (HR = 3.38, p = 0.021 and HR = 2.01, p = 0.150, respectively), emphasizing the complex interplay between patient characteristics and cancer outcomes [23]. The lack of significant differences in RFS across BMI groups within CRC patients also underscores the nuanced role of BMI in modulating disease progression post-surgery.

Our analysis supports the need for a more nuanced understanding of how BMI influences outcomes in specific cancer subtypes like CRC, where metabolic and systemic inflammation driven by obesity might affect tumor behavior differently compared to other cancers [24]. Future studies are warranted to explore these interactions further, potentially guiding more personalized treatment approaches based on patient BMI and tumor histology.

A least, according to literature, preoperative hypoalbuminemia is significantly associated with increased morbidity and mortality following gastrointestinal surgery, highlighting the importance of nutritional status as a predictor of surgical outcomes [25]. In our study, there was no significant difference in albumin levels between BMI groups (p = 0.690), indicating that nutritional status may not fully explain the observed survival differences. Further investigation into the role of BMI and nutritional status in surgical outcomes is warranted.

The limitations of this study revolve around the retrospective nature of the analysis. A significant challenge in our analysis arises from the heterogeneity of the patient group. The group of patients was quite heterogenous due to the various PM etiology and the two centres have different procedures and medical protocols. Prognostic outcomes are difficult to evaluate due to the inclusion of at least seven different etiologies of peritoneal malignant disease in various clinical scenarios (synchronous and metachronous metastases, preoperative chemotherapy, etc.). This fact generates a significant number of biases in the analysis and interpretation of the results. It's probably difficult to evaluate overall and disease-free survival by combining prognostically distinct disease entities, such as pseudomyxoma and peritoneal metastases from gastric or colorectal cancer. Additionally, we cannot definitively rule out the possibility that differences in survival outcomes may be influenced by the varied case mix between the two groups, specifically those with a BMI ≥ 25 compared to those with a BMI < 25 kg/m$^2$. Given these considerations, our findings should be interpreted with caution, recognizing that the observed associations might not solely reflect the influence of BMI but could also be a product of the complex interplay of multiple confounding factors. Prospective studies with more homogeneous patient populations and controlled variables are needed to clarify these relationships further and validate our findings.

## Conclusion

We have compared the oncological outcomes of two BMI groups with a follow-up period of 5 years in a European country. Morbidity at D30 was similar but readmission at D90 was higher in the high-BMI group. Our analysis identified BMI < 25 kg/m² is deleteriously associated with mortality. This study underscores the potential protective effect of higher BMI on overall survival in patients undergoing HIPEC, challenging traditional views on obesity in surgical outcomes. These findings suggest that BMI should be considered more carefully in preoperative evaluations and may influence treatment planning for peritoneal metastases. Future research should investigate the mechanisms behind the 'obesity paradox' to better tailor treatments for this population. Prospective studies with larger cohorts are essential to confirm these results and refine guidelines for managing patients with varying BMI levels undergoing HIPEC.

## Author contributions

**Conceptualization:** Anne-Cécile Ezanno, Catherine Arvieux, Fatah Tidadini.

**Data curation:** Anne-Cécile Ezanno, Olivier Poudevigne.

**Formal analysis:** Jean-Louis QUESADA.

**Funding acquisition:** Julio Abba, Brice Malgras, Bertrand Trilling, Pierre-Yves Sage.

**Investigation:** Fatah Tidadini.

**Methodology:** Jean-Louis QUESADA, Catherine Arvieux, Fatah Tidadini.

**Project administration:** Catherine Arvieux, Fatah Tidadini.

**Resources:** Brice Malgras.

**Supervision:** Anne-Cécile Ezanno, Juliette Fischer, Fatah Tidadini.

**Validation:** Anne-Cécile Ezanno.

**Writing – original draft:** Anne-Cécile Ezanno, Olivier Poudevigne, Brice Malgras, Fatah Tidadini.

**Writing – review & editing:** Jean-Louis QUESADA, Julio Abba, Bertrand Trilling, Pierre-Yves Sage, Juliette Fischer, Marc Pocard, Catherine Arvieux.

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
