## [Decision Letter · Decision Letter 0]

Dear Dr. Ezanno,

Thank you for submitting your manuscript to PLOS ONE. After careful consideration, we feel that it has merit but does not fully meet PLOS ONE’s publication criteria as it currently stands. Therefore, we invite you to submit a revised version of the manuscript that addresses the points raised during the review process.

We look forward to receiving your revised manuscript.

Kind regards,

Chong-Chi Chiu

Academic Editor

PLOS ONE

Journal Requirements:

https://www.ejso.com/article/S0748-7983(23)01569-X/abstract

In your revision ensure you cite all your sources (including your own works), and quote or rephrase any duplicated text outside the methods section. Further consideration is dependent on these concerns being addressed."""

Please provide additional details regarding participant consent. In the ethics statement in the Methods and online submission information, please ensure that you have specified (1) whether consent was informed and (2) what type you obtained (for instance, written or verbal, and if verbal, how it was documented and witnessed). If your study included minors, state whether you obtained consent from parents or guardians. If the need for consent was waived by the ethics committee, please include this information.

 3. In the online submission form, you indicated that [The data underlying the results presented in the study are available on request from the authors.].

Additional Editor Comments:

The reviewers have completed their initial review of your manuscript. Please revise your manuscript according to the suggestions.

Reviewers' comments:

Reviewer's Responses to Questions

**Comments to the Author**

1. Is the manuscript technically sound, and do the data support the conclusions?

Reviewer #1: No

Reviewer #2: Yes

Reviewer #3: Yes

2. Has the statistical analysis been performed appropriately and rigorously?

Reviewer #1: No

Reviewer #2: Yes

Reviewer #3: Yes

3. Have the authors made all data underlying the findings in their manuscript fully available?

Reviewer #1: Yes

Reviewer #2: Yes

Reviewer #3: Yes

4. Is the manuscript presented in an intelligible fashion and written in standard English?

Reviewer #1: Yes

Reviewer #2: Yes

Reviewer #3: Yes

Reviewer #1: This study aims to evaluate the impact of BMI in CRS+HIPEC. It includes all etiologies which makes for some very difficult interpretations. Otherwise, well-written article. My comments:

1: One major comment concerning the choice of etiologies. I think that the authors have to make a decision what to look at. If you are to only evaluate morbidity and postop mortality, I think it is reasonable to include all etiologies even though we know the dissection in PMP differs from gastric and colorectal. Nonetheless, a case can be made to include all CRS for a postop morbidity comparison. However, a postop survival comparison cannot be done with all these etiologies mixed in. For example in table 4, obviously gastric peritoneal metastases have a much worse survival than PMP or colorectal. The hazard ratio is high over 2 but due to small numbers doesn’t actually reach statistical significance. As such, etiologies are not included in the final multivariable analysis of survival, but obviously the histopathology will have a great impact on the multivariable model. I suggest either skipping the survival analyses completely or do a limited subgroup analysis removing all the small size etiologies, ie keep only colorectal, appendix, and PMP. Even if the univariate analysis is non-significant, the etiologies need to be included in the multivariable analysis anyways.

2: If you choose to focus solely on morbidity which I think is the cleanest option. It would be interesting to have a logistical regression model with grade 3+ yes/no endpoint or readmission within 90 days as endpoint. Clinically, the BMI and complication rates are mostly contemplated not so much survival. The clinically most important point is the surgical risk and complication rate.

3: I suggest testing different BMI cut-off as a sensitivity analysis. Do things change with higher BMI cut-off? Please provide a little table with Clavien-Dindo morbidity, Readmission 90D, and Reintervention 90D according to BMI <35 or 35+. This cut-off is usually used for when gastric bypass is offered. Perhaps the number of patients will be too small, but even if not including in the article, please provide it for review purposes. If the results are similar, you could just add a sentence in the results section mentioning that cut-off of BMI 35 did not change the outcomes.

4: Table 4 – This is almost a completely meaningless multivariable analysis. It is full of confounding variables as well as time dependant variables. This just does not suffice. For example, all the aggressive or poor prognostic etiologies (gastric and advanced colorectal) receive neoadjuvant therapy while the very good prognostic ovary and PMP usually do not. Thus, neoadjuvant therapy looks to have a terrible hazard ratio which is clearly misguiding. In other words, the neoadjuvant variable is directly affected by and determined by the etiologi variable. You have repeat patients as well. This group is by definition a selected group from a selected group compared to those doing their first CRS+HIPEC. You have adjuvant therapy included which also is determined by etiologi and postop complications. Intraperitoneal chemotherapy is also completely etiologi dependent. Mitomycin C is the standard drug for PMP which has a super good prognosis which makes it misguided. Since etiologi is not apart of the multivariable analysis, these differences are not adjusted correctly. This multivariable model is just not feasible or methodologically sound. See comment 1. I am not sure a Cox model for overall survival including all etiologies is feasible. Besides the limited number of patients, the other variables such as neoadjuvant and adjuvant chemo is never offered low grade PMP. So when you have this situation with 0 patients in some of the subgroups, the multivariable analyses tend to fail. In principle, all the baseline variables chosen need to be possible to fall into the included subgroups. If this is not the case, the model becomes problematic.

5: Basically, the same reasoning for Table 5 as it is for Table 4. I would suggest discussing with a statistician. My suggestion would be to skip the Cox regression modelling for overall and recurrence free survival and focus on the postop morbidity (and readmission) instead.

Thank you for giving me the opportunity to review this manuscript!

Reviewer #2: The study explores the influence of body mass index (BMI) on surgical and oncological outcomes in patients undergoing cytoreductive surgery with hyperthermic intraperitoneal chemotherapy (HIPEC) for peritoneal metastases. The relevance of this research lies in the rising rates of obesity and overweight in Europe and the need to understand better how BMI impacts survival and postoperative complications in this context of intensive treatment.

This is a retrospective observational study conducted at two HIPEC expert centers in France from 2017 to 2021. Patients are divided into two groups based on BMI (<25 and ≥25). The primary endpoints are 30-day morbidity, overall survival (OS), and recurrence-free survival (RFS). The statistical analysis uses a Cox model to adjust for confounding variables, including sex, neoadjuvant chemotherapy, and the HIPEC techniques used (open or closed).

The results show that:

- Overall survival is significantly higher for patients with BMI≥25. The "obesity paradox" is suggested to explain this paradoxical association where a higher BMI is linked to better survival.

- 30-day morbidity is similar between both groups, although the high-BMI group showed an increased rate of digestive fistulas and 90-day readmissions.

- Factors such as male sex, neoadjuvant chemotherapy, and the use of mitomycin C in HIPEC also influence survival.

The article highlights an inverse association between BMI and postoperative mortality, supporting the "obesity paradox." This phenomenon is intriguing but remains debated, particularly in oncology. However, the study presents certain limitations that complicate interpretation:

- Heterogeneity of patients: Patients have peritoneal metastases from various origins (colorectal, pseudomyxoma, ovarian, etc.), which could introduce bias. Survival outcomes may vary depending on cancer type, making it difficult to attribute a specific effect of BMI on survival.

- Retrospective design: The retrospective nature of the study limits the robustness of conclusions. Data are subject to selection and reporting bias, and analysis relies on the quality of medical records.

- Lack of data on other confounding factors: Although the study adjusts for some factors like sex and HIPEC technique, other variables (e.g., precise nutritional status, unreported comorbidities) might also affect the outcomes.

The study concludes that a BMI≥25 might be protective for the overall survival of patients undergoing HIPEC treatment, but it calls for further studies to confirm these findings and explore additional factors in the long term. It also highlights the need for a better understanding of the obesity paradox in surgical oncology.

The study presents a relevant analysis on a clinically important topic. However, methodological biases and limitations reduce the strength of the conclusions. A prospective design or finer stratification of patient groups by pathology could enhance the validity of the results.

Reviewer #3: This retrospective two-center study examined the impact of body mass index (BMI) on surgical and oncological outcomes in 144 patients who underwent complete cytoreductive surgery with HIPEC for peritoneal metastases between 2017 and 2021. Patients with BMI<25 showed significantly poorer overall survival compared to those with BMI≥25, although recurrence-free survival was similar between groups. While 30-day complication rates were comparable between groups, patients with BMI≥25 experienced more digestive fistulas and higher 90-day readmission rates. The study identified several significant predictors of outcomes, including BMI<25 and male sex as risk factors for poorer overall survival, while pseudomyxoma histology emerged as a protective factor. Overall, the study is interesting and the manuscript is well-written. Thus, I just have one minor concern.

1. May add the subgroup analysis according to BMI to validate the significant association between BMI and clinical outcomes of the patients.

2.Please define the comorbidities, such as cardiopathy, mini-stroke, renal failure, hepatopathy in the table 1.

**Do you want your identity to be public for this peer review?** For information about this choice, including consent withdrawal, please see our Privacy Policy

Reviewer #1: No

Reviewer #2: No

Reviewer #3: No

---

## [Author Response · Author response to Decision Letter 1]

17 Dec 2024

Dear Editor,

We are resubmitting a revised version of our manuscript “Impact of body mass index on surgical and oncological outcomes after Hyperthermic Intraperitoneal Chemotherapy” that we hope is now suitable for publication in the PlosOne.

We thank the editor and the reviewers for their comments and suggestions, which have helped us to improve the article. We provide point-by-point responses to the comments of the reviewer. We include a “clean” version and a “marked” version.

We hope that our manuscript now meets the high standards of PlosOne, can contribute to the literature on this new technique, to the debate about its pros and cons, and the best trial design criteria to evaluate it further.

Yours sincerely,

Anne-Cécile Ezanno , on behalf of all authors.

---

## [Decision Letter · Decision Letter 1]

Dear Dr. Ezanno,

Thank you for submitting your manuscript to PLOS ONE. After careful consideration, we feel that it has merit but does not fully meet PLOS ONE’s publication criteria as it currently stands. Therefore, we invite you to submit a revised version of the manuscript that addresses the points raised during the review process.

Please submit your revised manuscript by Apr 23 2025 11:59PM. If you will need more time than this to complete your revisions, please reply to this message or contact the journal office at plosone@plos.org . A rebuttal letter that responds to each point raised by the academic editor and reviewer(s). You should upload this letter as a separate file labeled 'Response to Reviewers'.A marked-up copy of your manuscript that highlights changes made to the original version. You should upload this as a separate file labeled 'Revised Manuscript with Track Changes'.An unmarked version of your revised paper without tracked changes. You should upload this as a separate file labeled 'Manuscript'.

We look forward to receiving your revised manuscript.

Kind regards,

Chong-Chi Chiu

Academic Editor

PLOS ONE

**Journal Requirements:**

**Additional Editor Comments:**

The reviewers have finished the review. Please finish the minor revision according to the reviewer 4.

Reviewers' comments:

Reviewer's Responses to Questions

**Comments to the Author**

Reviewer #3: All comments have been addressed

Reviewer #4: All comments have been addressed

2. Is the manuscript technically sound, and do the data support the conclusions?

Reviewer #3: Yes

Reviewer #4: Partly

3. Has the statistical analysis been performed appropriately and rigorously?

Reviewer #3: Yes

Reviewer #4: Yes

4. Have the authors made all data underlying the findings in their manuscript fully available?

Reviewer #3: Yes

Reviewer #4: Yes

5. Is the manuscript presented in an intelligible fashion and written in standard English?

Reviewer #3: Yes

Reviewer #4: Yes

**Reviewer #3: ** The authors had addressed well all my concern, so I have no more suggestion for this revision. Congratulation

**Reviewer #4:**  I read with interest the manuscript “Impact of body mass index on surgical and oncological outcomes after Hyperthermic Intraperitoneal Chemotherapy (HIPEC)”. The paper faced on an interesting topic, the evaluation of BMI in patients treated with CRS-HIPEC as risk factor for postoperative outcome and oncological long term results. The text is clear and easy to follow. However, there are important drawbacks of this work.

1-The number of patients is too low for draw any reliable conclusion

2-The impact of BMI on morbidity and mortality seems null. This is quite surprising. How explain the authors this result?

3-A higher incidence of intestinal fistula is reported. How do the authors explain this result?

4-OS is higher in obese patients. How do the authors explain this result?

However, the main problem of the manuscript is the discussion where none of the results of the study have been discussed or interpreted. Moreover, the authors didn’t highlight the importance of their results. Why are the results important? How can it change the clinical practice or set the ground of future studies?

**Do you want your identity to be public for this peer review?** For information about this choice, including consent withdrawal, please see our Privacy Policy

Reviewer #3: No

Reviewer #4: No

---

## [Author Response · Author response to Decision Letter 2]

12 Mar 2025

Dear Editor,

We are pleased to resubmit our revised manuscript for consideration in PLOS ONE. The manuscript has been thoroughly revised to address the insightful comments and suggestions provided by the reviewers.

We have also included a detailed point-by-point response to the reviewers' comments with this resubmission for your consideration.

We hope that our revised manuscript now meets the high standards of PLOS ONE and that it will be considered suitable for publication. We appreciate the opportunity to revise our work and thank the editorial team and reviewers for their constructive and valuable input.

We look forward to your feedback and are happy to make further adjustments if necessary.

Sincerely

---

## [Decision Letter · Decision Letter 2]

Impact of body mass index on surgical and oncological outcomes after Hyperthermic Intraperitoneal Chemotherapy (HIPEC)

PONE-D-24-34218R2

Dear Dr. Ezanno,

We’re pleased to inform you that your manuscript has been judged scientifically suitable for publication and will be formally accepted for publication once it meets all outstanding technical requirements.

Kind regards,

Chong-Chi Chiu

Academic Editor

PLOS ONE

Additional Editor Comments (optional):

The authors have revised the manuscript according to the reviewers' suggestions. The quality of the revised manuscript is acceptable.

Reviewers' comments:

Reviewer's Responses to Questions

**Comments to the Author**

Reviewer #3: All comments have been addressed

2. Is the manuscript technically sound, and do the data support the conclusions?

Reviewer #3: Yes

3. Has the statistical analysis been performed appropriately and rigorously?

Reviewer #3: Yes

4. Have the authors made all data underlying the findings in their manuscript fully available?

Reviewer #3: Yes

5. Is the manuscript presented in an intelligible fashion and written in standard English?

Reviewer #3: Yes

Reviewer #3: The authors response well for this revision, so I have no more suggestions and recommend accept for this article.

**Do you want your identity to be public for this peer review?** For information about this choice, including consent withdrawal, please see our Privacy Policy

Reviewer #3: No

---

## [Editor Report · Acceptance letter]

PONE-D-24-34218R2

PLOS ONE

Dear Dr. Ezanno,

I'm pleased to inform you that your manuscript has been deemed suitable for publication in PLOS ONE. Congratulations! Your manuscript is now being handed over to our production team.

Kind regards,

on behalf of

Professor Chong-Chi Chiu

Academic Editor

PLOS ONE